materials science

polysilazane, epoxy, polymer, foam

**Authors for correspondence:**
Yongming Luo
e-mail: luoym@iccas.ac.cn
Caihong Xu
e-mail: caihong@iccas.ac.cn
Jianling Zhao
e-mail: zhaojl@hebut.edu.cn

This article has been edited by the Royal Society of Chemistry, including the commissioning, peer review process and editorial aspects up to the point of acceptance.

# Polysilazane as a new foaming agent to prepare high-strength, low-density epoxy foam

Yucheng Chang[1], Yongming Luo[2], Caihong Xu[2] and Jianling Zhao[1]

[1]School of Material Science and Engineering, Hebei University of Technology, Tianjin 300130, People's Republic of China
[2]Institute of Chemistry, Chinese Academy of Sciences, Beijing 100190, People's Republic of China

(iD) YL, 0000-0002-4709-0846

Polysilazane (PSN2) was used as chemical foaming agent to prepare epoxy foam for the first time. Using the new foaming agent, epoxy foams with high strength and low density were successfully prepared. The density of epoxy foam can be tuned from 0.321 to 0.151 g cm$^{-3}$ by changing the content of PSN2 from 2.50 to 7.50 wt%, with the compressive strength varied from 7.39 to 1.25 MPa. The morphology, porosity, mechanical property, thermal conductivity and adhesive property of foams with different polysilazane content were investigated. Besides, the effect of polyamine curing agent and surfactant on foams was also investigated to optimize the as-prepared epoxy foam.

## 1. Introduction

Porous polymers have been found to be enabling technological components in various engineering applications such as adsorption, separation, filtration, sensors, drug delivery and energy storages [1–5]. Various polymer foams, including epoxy, phenolic, cyanate ester, polyimide, polyurethane, polyester, silicone, nylon, polybutadiene, polypropylene, etc., have been reported [6–15]. Among them, epoxy foam is being widely used in sandwich composites and thermal protection systems of rockets to protect the substructures due to its excellent adhesion and mechanical properties. Epoxy foam is a rigid polyfoam, which can be prepared by adding curing agents and foaming agents into epoxy resin matrix. Compared with microspheres/bead foaming [16–20] and physical foaming [21–24], low-density epoxy foam can be easily prepared because the process that is used in chemical foaming agents is simpler. However, there are few

reports on using the simple process to prepare epoxy foam due to the disadvantage of using traditional chemical foaming agents in other processes. Aluminium powder−NaOH and $NaHCO_3$ were reported as chemical foaming agents of epoxy resin [25,26]. The compatibility of these generally used chemical foaming agents with epoxy resin is not very good, resulting in quite inhomogeneous and irregular microcells. After foaming, the residue of the above chemical foaming agent remains in the matrix and affects the properties of foam. Wang *et al.* [27,28] carried out a constructive work on preparing epoxy foam with 3,7-dinitroso-1,3,5,7-tetraazobicyclononane as the chemical foaming agent. They successfully reduced the microcell size of epoxy foam by the limited foaming process, but the foam-forming process requires high temperature (110°C) and more time (3 h). Stefani *et al.* [29−31] used siloxane as the chemical foaming agent to prepare epoxy foam via the reaction between siloxane and polyamine hardener, but the obtained foam showed poor mechanical property. To achieve high-performance epoxy foam with the simple technique, it is necessary to develop a new chemical foaming method and a foaming agent.

Polysilazane is a well-known polymeric precursor for silicon nitride ceramic and it has found various applications in the fields of high temperature-resistant adhesive, coating, water-repellent layer and silicon nitride fibre [32−37]. It also has been successfully used to modify other organic oligomers and polymers, for example, poly(methyl methacrylate), polyacrylonitrile, unsaturated polyester and aniline formaldehyde resin and epoxy resin [38−43]. However, there are few reports about modifying epoxy resin with polysilazane. In a US Patent, Lukacs used polysilazane to modify polyamine, the hardener for epoxy resin. The modified hardeners enhanced high-temperature properties of cured epoxy resin [43]. Besides the expected reaction between Si−H and −$NH_2$, they found some side reactions which complicated the modifying process. The Si−N bonds of some polysilazanes can react with active free hydroxyl groups generated in the oxirane ring opening process to form Si−O−C and ≡Si−$NH_2$. The ≡Si−$NH_2$ can further react with other hydroxyl groups to generate $NH_3$. Such tricky side reactions in that research suggested to us that polysilazane is a potential chemical foaming agent for epoxy resin.

In this work, polysilazane was used as a chemical foaming agent to prepare epoxy foam for the first time. The epoxy foams with high strength, low density were successfully prepared, the density of which can be tuned from 0.321 to 0.151 g cm$^{-3}$ by changing the content of polysilazane from 2.50 to 7.50 wt%, with their compressive strength varying from 7.39 to 1.25 MPa. The influences of polysilazane polyamine hardener and surfactant content on the microstructure, apparent density, porosity and mechanical property of the formed foam were investigated in detail. Furthermore, thermal insulation property and adhesive property of foams were also characterized.

# 2. Material and methods

## 2.1. Materials

Commercial bisphenol-A type epoxy resin (E-51, 0.52−0.54 eq/100 g, Nantong Xingchen Synthetic Material Co. Ltd, China) was used as matrix resin. Triethylenetetramine (TETA, 67%, Aladdin Industrial Corporation, China) was used as a hardener. Polysilazane (PSN2, 600 g mol$^{-1}$, 72.02 mPa s$^{-1}$), which contains Si−H functional group and synthesized via the aminolysis of chlorosilane according to our previous work, was used as a foaming agent. In addition, emulsifier OP-10 (Aladdin Industrial Corporation, China) was added into the system as a surfactant to stabilize the foam.

## 2.2. Processing of the foam

Schematic diagram of the foam preparation process is shown in figure 1. The foam was prepared in a two-step batch process [22]. The uniform mixture of E-51, TETA and OP-10 in a beaker was first precured for 30 min to gain proper viscosity. After the addition of PSN2, the precured mixture was well stirred. Then the mixture was transferred carefully into a mould for foaming and curing at room temperature for 3 h. Subsequently, the foamed sample was transferred into an oven to further foam and cure at 80°C for 2 h. Finally, epoxy foam was obtained after cooling and demoulding. For formulation optimization, the weights of the PSN2, TETA and OP-10 were changed while keeping the weights of other components constant. The used contents of PSN2, TETA and OP-10 are listed in table 1.

The chemical formula of PSN2 is shown in figure 2. During the foaming process, $NH_3$ and $H_2$ were generated via two parallel processes, as shown in figure 2. The Si−N bonds in polysilazanes react with hydroxyl groups generated by the oxirane ring opening reaction to form Si−O−C and ≡Si−$NH_2$. The

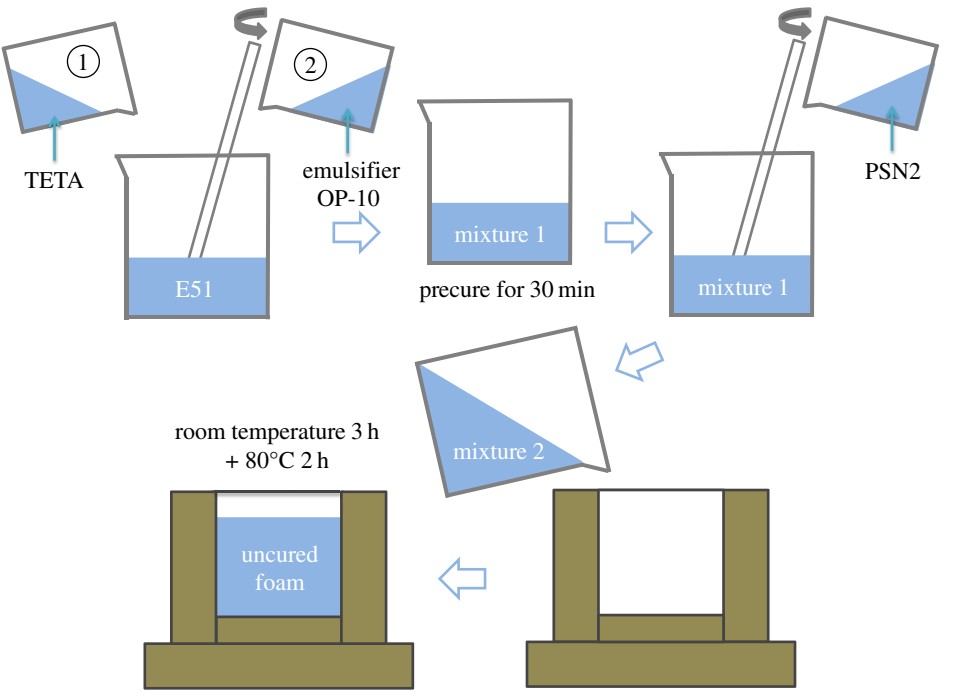

**Figure 1.** Schematic diagram of foam preparation process.

**Figure 2.** Chemical formula of PSN2 and reactions in foaming process.

**Table 1.** Content of components in epoxy foam.

| no. 1 | PSN2 content (wt%) | | | | | TETA content (wt%) | OP-10 content (wt%) |
|---|---|---|---|---|---|---|---|
| | 2.5 | 3.75 | 5 | 6.25 | 7.5 | 10 | 1 |
| no. 2 | TETA content (wt%) | | | | | PSN2 content (wt%) | OP-10 content (wt%) |
| | 10 | 11.25 | 12.5 | 13.75 | 15 | 5 | 1 |
| no. 3 | OP-10 content (wt%) | | | | | PSN2 content (wt%) | TETA content (wt%) |
| | 1 | 2 | 3 | 4 | 5 | 2.5 | 10 |

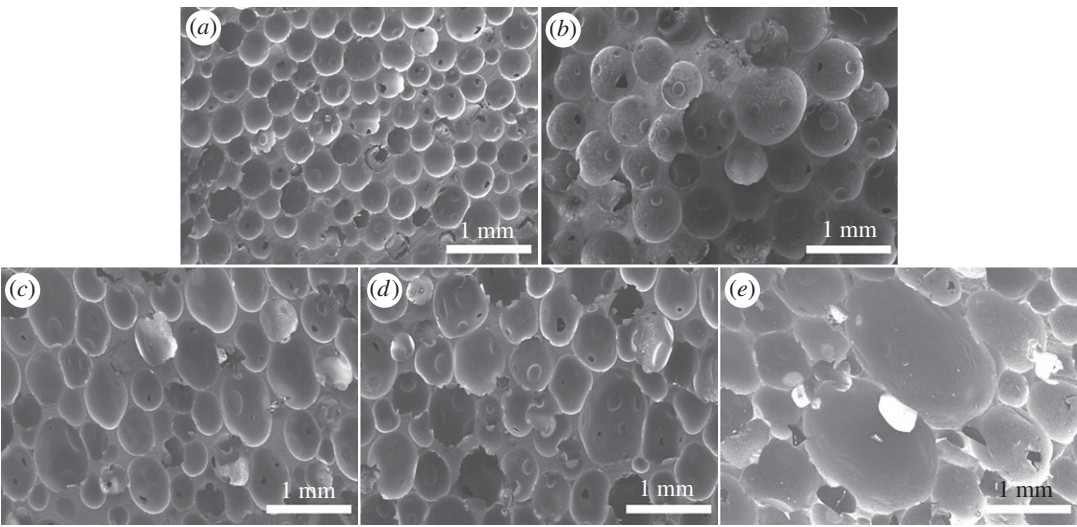

**Figure 3.** SEM micrographs of epoxy foams with (*a*) 2.50 wt% PSN2; (*b*) 3.75 wt% PSN2; (*c*) 5.00 wt% PSN2; (*d*) 6.25 wt% PSN2; (*e*) 7.50 wt% PSN2.

$\equiv$Si–NH$_2$ groups then further react with other hydroxyl groups and release NH$_3$ (process 1). The Si–H groups in polysilazanes also can react with –NH$_2$ of TETA to produce H$_2$. The release of NH$_3$ and H$_2$ resulted in the formation of porous structure in the foam. In the meantime, the chemical structure unit of PSN2 was introduced into the epoxy–polyamine system effectively, which was quite different from the reported mechanism of other epoxy foaming agents.

## 2.3. Characterization

The morphology of epoxy foam was observed with a scanning electron microscope (SEM, Hitachi S-4800, Japan). The apparent density was calculated from weight, height and diameter of cylindrical foam sample according to the following formula:

$$\rho = \frac{4m}{\pi \, d^2 \, h},\tag{2.1}$$

in which $\rho$ is the apparent density and $m$, $h$ and $d$ represent weight, height and diameter, respectively. The porosity of epoxy foam was calculated from apparent density and the density of dense epoxy bulk (1.122 g cm$^{-3}$) with the following formula (2.2):

$$\theta = \left(\frac{1-\rho}{\rho_s}\right) \times 100\%,\tag{2.2}$$

in which $\theta$ is the porosity and $\rho$ and $\rho_s$ represent apparent density and the density of dense epoxy bulk, respectively. The mechanical properties, including compressive strength, flexural strength and tensile strength, were characterized using a universal material testing machine (Instron 5567, USA) according to GBT8813-2008, GBT8812-2007 and GB9641-1988. The thermal conductivity was tested with a thermal conductivity analyser (C-THERM TCI, Canada) at room temperature. The shear strength of epoxy foam adhesive was characterized using a universal material testing machine (Instron 5567, USA) according to GJB1480A-2005.

# 3. Results and discussion

## 3.1. Effect of PSN2 on foam

Figure 3 shows the morphology of epoxy foams using the same content of TETA and OP-10 (10.00 and 1 wt%, respectively) but different amount of PSN2. As we can see, the size of microcells increases significantly with increasing PSN2 content because the amount of gas generated in reaction increases. In this case, the microcell grew faster under the influence of gas pressure, resulting in larger cell size. Figure 3*a*–*c* exhibits that the homogeneity of microcells is relatively fine as the content of PSN2 increases from 2.50 to 5.00 wt%. However, further content increase in PSN2 to 6.25 and 7.50 wt%

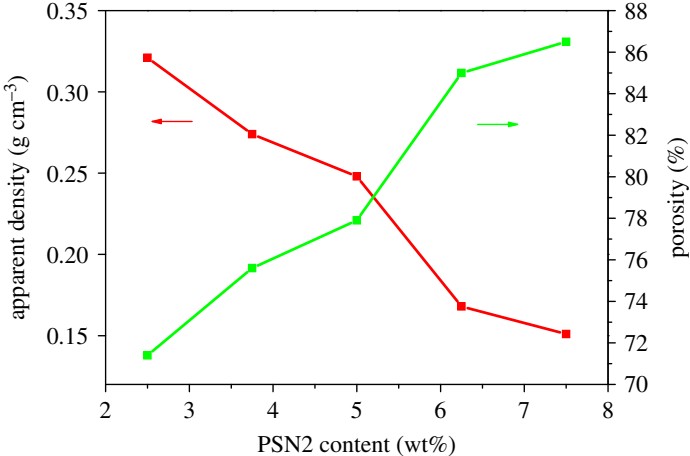

**Figure 4.** Apparent density and porosity of epoxy foams with different PSN2 content.

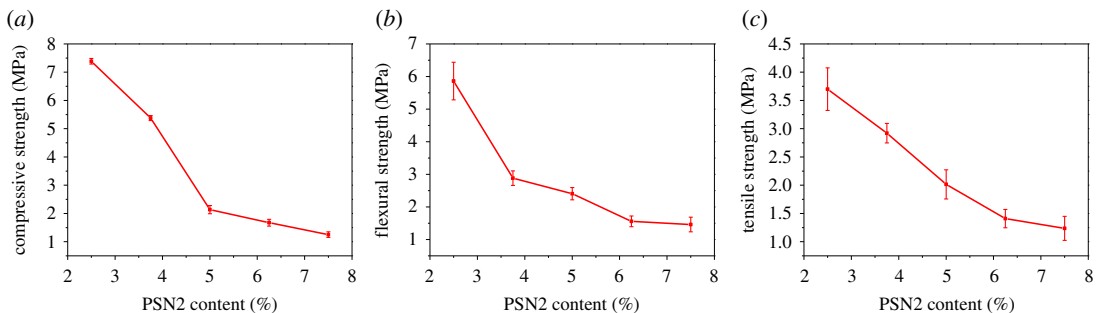

**Figure 5.** (*a*) Compressive strength; (*b*) flexural strength; (*c*) tensile strength of epoxy foams with different PSN2 content.

resulted in the decrease in size homogeneity and shape regularity as the microcells further expand, especially for the foam with the PSN2 content of 7.50 wt% (figure 3*d,e*). Such phenomenon is due to the fast generation of gas when the PSN2 is excessive. The epoxy matrix cannot get enough strength via curing to support foaming, so the microcells deform and merge during expanding.

Figure 4 shows the effect of PSN2 content on apparent density and porosity of epoxy foam. It can be seen that by changing the content of PSN2, densities of the foams can be tuned in a wide range. Along with the rise of PSN2 content, the apparent density of epoxy foam decreases from 0.321 to 0.151 g cm$^{-3}$, while the porosity increases from 71.4 to 86.5%. But as the content of PSN2 increases from 6.25 to 7.50 wt%, the decrease in density and the increase in porosity are both not so obvious while the deterioration of morphology is serious as shown in figure 3*e*.

The mechanical property of epoxy foams with different PSN2 content is shown in figure 5. It is found that as the density of foam decreases with the rising of PSN2 content, the mechanical property also decreases. The compressive strengths of foams with PSN2 content of 2.50, 3.75, 5.00, 6.25 and 7.50 wt% are 7.39, 5.38, 2.14, 1.67 and 1.25 MPa, respectively, as shown in figure 5*a*, which are quite high compared to those of previously reported epoxy foams with similar density [6,25,29,44–46]. Such improvement demonstrates that PSN2 was not only a foaming agent but also a modifier of the as-prepared epoxy foam. Figure 5*b,c* shows the flexural strength and tensile strength of foams with different PSN2 contents. The flexural strength and tensile strength decrease from 5.86 and 3.70 MPa to 1.46 and 1.23 MPa, respectively, along with the density decrease from 0.321 to 0.151 g cm$^{-3}$ caused by the rise of PSN2 content. There are few reports about these two properties of epoxy foam; however, they are pretty important for practical application. So, the characterization of them is a good reference for future research and application.

The thermal insulation performance of such epoxy foam was also studied. Figure 6 shows the thermal conductivity of foams with different PSN2 contents. The specific thermal conductivity values of the foams with PSN2 content of 2.50, 3.75, 5.00, 6.25 and 7.50 wt% are 0.070, 0.068 0.065, 0.053 and 0.059 W (m K)$^{-1}$, respectively, which are all much lower than that of dense epoxy block (0.133 W (m K)$^{-1}$). Such phenomenon can be explained by the special heat transfer mechanism of foam

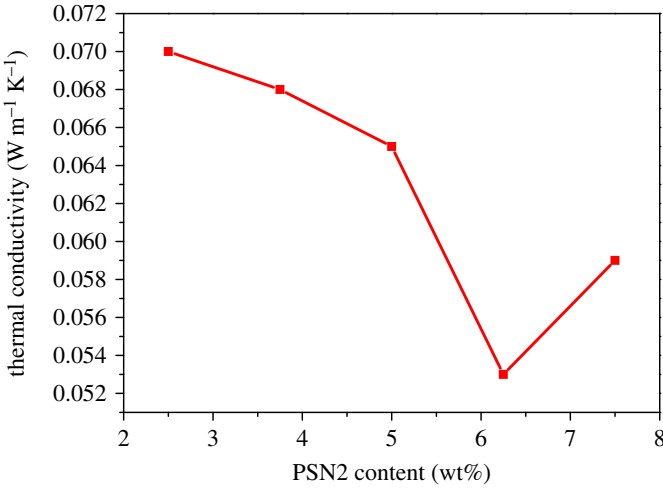

**Figure 6.** Thermal conductivities of epoxy foams with different PSN2 content.

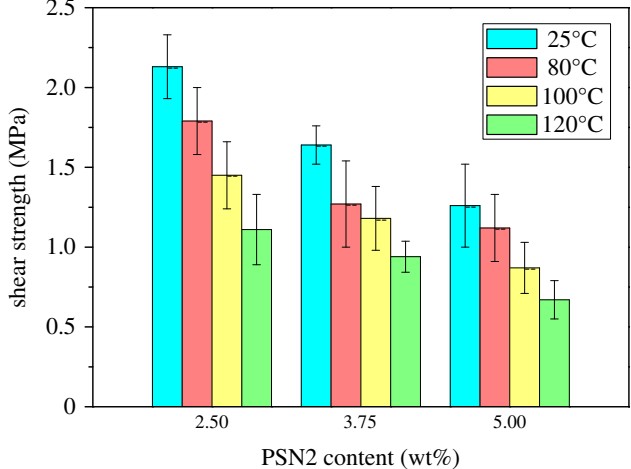

**Figure 7.** Shear strength of foam adhered concentric tubes at different temperature.

materials [47]. There are two thermal transfer modes when the heat transmits through microcells. One is the heat transfer through the wall of microcells, whose thermal transfer distance is much longer than that of dense block. The other is the heat transfer through the gas in microcells, which is much slower because the thermal conductivity of gas is much lower than that of solid matrix. From figure 6, we can see that the thermal conductivity decreases as the PSN2 content increases from 2.50 to 6.25 wt%. The rise of PSN2 content enlarges the gas volume in the foam and causes the proportional increase in the second thermal transfer mode, thus improving the thermal insulation performance of foam. But as the PSN2 content rises from 6.25 to 7.50 wt%, the thermal conductivity increases instead of decreasing. The violent expansion and merge of microcells made the surface area of microcells decrease and thus shortened the thermal transfer distance, while the density decrease and porosity increase caused by the rise of PSN2 content are quite low. Therefore, the thermal conductivity increases. Even so, the thermal insulation performance of the foam with the PSN2 content of 7.50 wt% is still better than that of foam with the PSN2 content of 5.00 wt%, because of the large difference of density. In general, the as-prepared epoxy foam is a potential thermal insulation material, and the thermal conductivity of such foam can be tuned by changing the content of PSN2.

As we can know from the above, the foams with 2.50, 3.75 and 5.00 wt% PSN2 exhibited relatively good morphology and mechanical property. So, their corresponding unfoamed precursors were further used as foam adhesive in the adhesion of aluminium alloy concentric tubes. The shear strengths of the epoxy foam adhesive were tested at different temperatures and the result is shown in figure 7. The shear strength decreases with increasing PSN2 content, which is similar to the change trend of the mechanical properties of foam itself due to the density reduction. Besides, it is found that

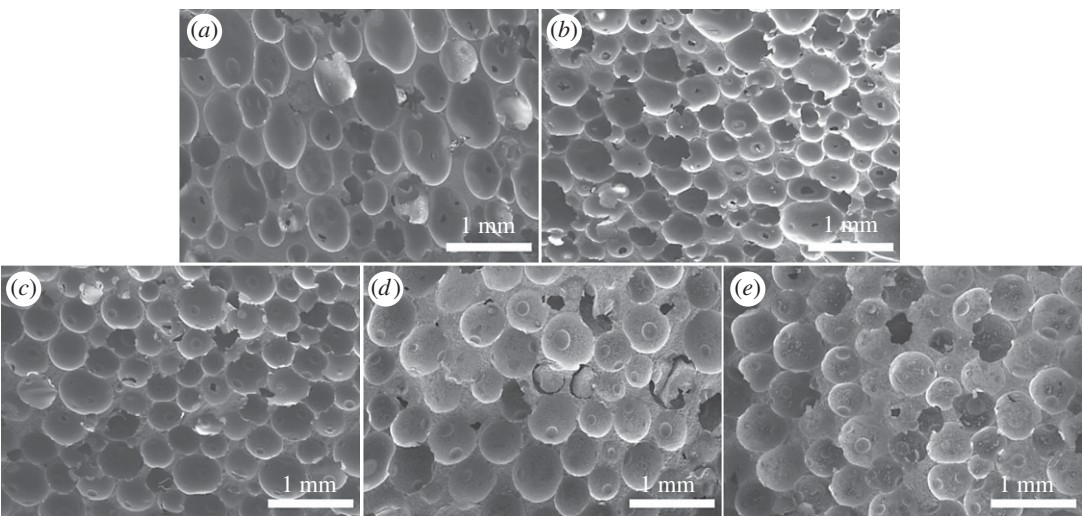

**Figure 8.** SEM micrographs of epoxy foams with (*a*) 10.00 wt% TETA; (*b*) 11.25 wt% TETA; (*c*) 12.50 wt% TETA; (*d*) 13.75 wt% TETA; (*e*) 15.00 wt% TETA.

**Table 2.** Apparent density and porosity of epoxy foams with different TETA content.

| TETA content (wt%) | 10 | 11.25 | 12.5 | 13.75 | 15 |
|---|---|---|---|---|---|
| apparent density (g cm$^{-3}$) | 0.248 | 0.251 | 0.256 | 0.262 | 0.267 |
| porosity (%) | 77.9 | 77.6 | 77.2 | 76.6 | 76.2 |

the shear strength decreases with the increase in testing temperature. The shear strength tested at 120°C is only approximately half of that at room temperature. However, at 80 and 100°C, the foam adhesive still can maintain relatively high shear strength. In general, the epoxy foam we developed can be used as low-density foam adhesive from room temperature to 100°C.

## 3.2. Effect of TETA on foam

The effect of curing agent content on the morphology and mechanical property of the foams was also investigated. As can be seen from figure 8, keeping the contents of PSN2 and OP-10 for all testing samples at 5.00 wt% and 1 wt%, respectively, the morphology becomes better with the TETA content rising from 10.00 to 15.00 wt%. The size of microcells decreases along with the increase in TETA content. In the meantime, the homogeneity of microcells improves. The shape of microcells becomes more regular. The morphology improves because the increase in TETA content accelerates the curing process. The foam framework with more TETA can gain greater strength at the same foaming stage, which prevents the overexpansion and deformation of microcells. Hence, the morphology of epoxy foam can be optimized via adjusting the content of TETA.

Apparent density and porosity of epoxy foams with different TETA content are shown in table 2. It is found that the apparent density of foam slightly increases as the content of TETA rises from 10.00 to 15.00 wt% but still under 0.27 g cm$^{-3}$ due to the acceleration of curing. The porosity decreases but still over 76%. The mechanical property of epoxy foams with different TETA content is shown in figure 9. We can see that the mechanical property improves as the TETA content rises from 10.00 to 15.00 wt%, and the sample with TETA content of 15.00 wt% has its compressive strength, flexural strength and tensile strength up to 4.44, 4.08 and 3.52 MPa, respectively. Such a phenomenon can be attributed to the density increase and the morphology improvement as mentioned above. The increase in TETA content not only contributes to the morphology improvement but also reinforces the framework of foam via improving the curing degree. Therefore, increasing the content of curing agent is an effective way to optimize epoxy foam. However, the increase of content is limited, because excessive amount of curing agent will make the curing process too fast to have enough effective foaming time. The apparent density of foam will be over 0.27 g cm$^{-3}$ when the content of TETA is higher than 15.00 wt%. Besides, when the content of TETA is 8.75 wt% or less, the epoxy cannot be cured effectively.

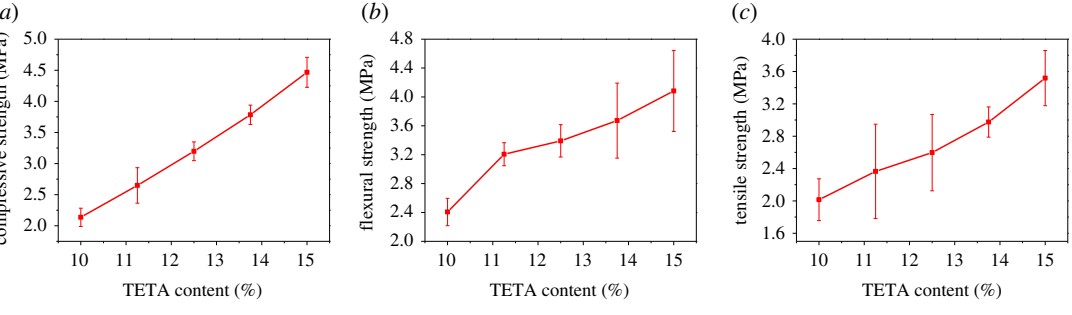

**Figure 9.** (*a*) Compressive strength; (*b*) flexural strength; (*c*) tensile strength of epoxy foams with different TETA content.

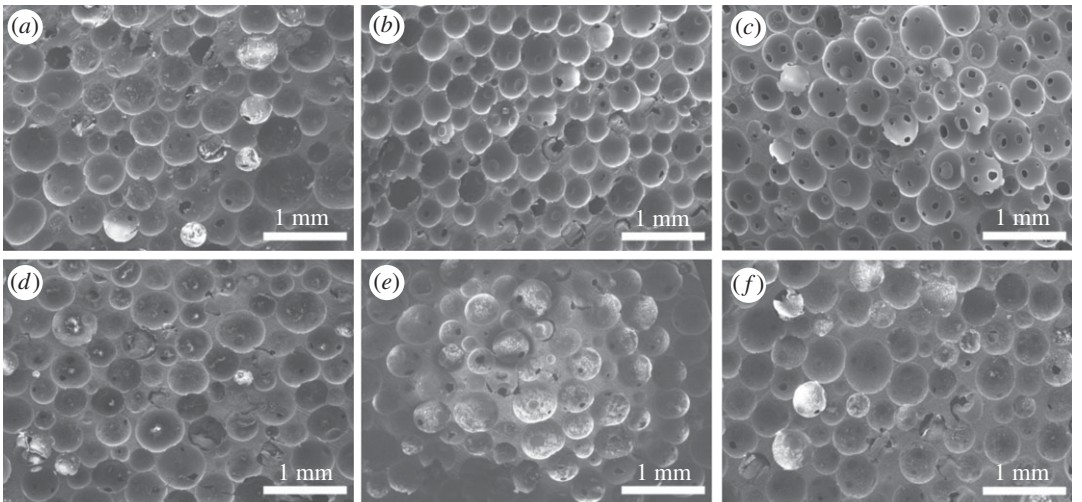

**Figure 10.** SEM micrographs of epoxy foams with (*a*) 0 wt% OP-10; (*b*) 1 wt% OP-10; (*c*) 2 wt% OP-10; (*d*) 3 wt% OP-10; (*e*) 4 wt% OP-10; (*f*) 5 wt% OP-10.

## 3.3. Effect of OP-10 on foam

OP-10 is used as a surfactant in the preparation of the epoxy foam. Though the content of OP-10 is quite low, it had a significant effect on epoxy foam. The surfactant can affect epoxy foam in three ways. First, it can homogenize the size of microcells via increasing the compatibility among polar PSN2, polar epoxy matrix and non-polar curing agent. Second, it can facilitate the formation of microcells via lowering the surface tension of the mixture. Finally, it can increase the surface viscosity and elasticity of the microcells thus can prevent the microcells from bursting. Therefore, the effect of OP-10 on the morphology and mechanical property of the epoxy foam was also investigated in this research. The content of OP-10 in foam varied from 0 to 5 wt% as the content of PSN2 (2.50 wt%) and TETA (10.00 wt%) remained unchanged. Figure 10 shows the morphology of foams with different OP-10 contents. Comparing figure 10*a* with figure 10*b*,*c*, it can be found that the size of microcells becomes more homogeneous and the shape of microcells becomes more regular after the addition of OP-10. But when the OP-10 content is up to 3 and 4 wt%, the microcells begin to deform as figure 10*c*,*d* shows. When the OP-10 content is further raised to 5 wt%, the homogeneity of microcells begins to decline. Such phenomenon may be attributed to the strength reduction in bubble wall caused by the addition of OP-10 during the curing and foaming process. Therefore, judging from the morphology of foams, the optimum content of OP-10 is 1–2 wt%. Both the microcell homogeneity and regularity of foam using optimized content of OP-10 are better than those of the others.

Figure 11 shows the compressive strength of epoxy foams with different OP-10 content. Though the addition of OP-10 stabilizes the morphology effectively, it has adverse influence on the compressive strength of foams. The compressive strength decreases obviously along with the rise of OP-10 content, which is mainly attributed to the decrease of apparent density as shown in table 3. Besides, strength reduction of foam framework caused by the addition of OP-10 is also an important reason. Considering both stability and strength, the epoxy foam can achieve fine compressive property with the OP-10 of 1 wt%. The compressive strength is relatively high while the stability is fine.

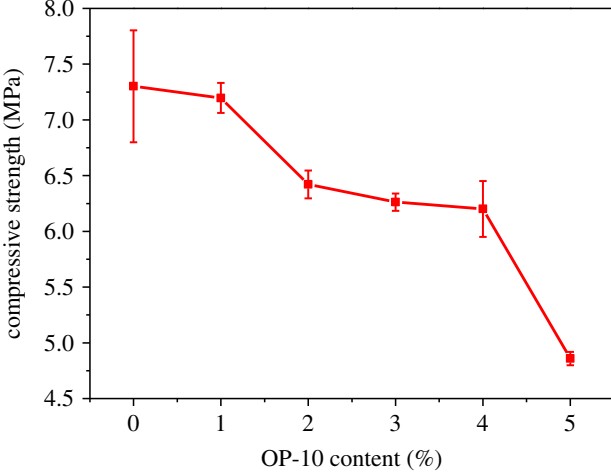

**Figure 11.** Compressive strength of epoxy foams with different OP-10 content.

**Table 3.** Apparent density and porosity of epoxy foams with different OP-10 content.

| OP-10 content (wt%) | 0 | 1 | 2 | 3 | 4 | 5 |
|---|---|---|---|---|---|---|
| apparent density (g cm$^{-3}$) | 0.332 | 0.321 | 0.314 | 0.302 | 0.288 | 0.261 |
| porosity (%) | 70.4 | 71.4 | 72 | 73.1 | 74.3 | 76.7 |

# 4. Conclusion

Epoxy foams were successfully prepared using PSN2 as a chemical foaming agent for the first time. The density and porosity of epoxy foam can be tuned from 0.321 g cm$^{-3}$ and 71.4% to 0.151 g cm$^{-3}$ and 86.5%, respectively, by changing the content of PSN2 from 2.50 to 7.50 wt%. As the result exhibits, the morphology, mechanical property, thermal conductivity and adhesive property of epoxy foam change along with the density and porosity change caused by the change of PSN2 content. Especially, the compressive strength of the as-prepared foam is quite good compared with other epoxy foams that have a similar density, which demonstrates that PSN2 is not only a foaming agent but also a modifier of the epoxy foam. In addition, both the morphology and mechanical property of the foam can be optimized by choosing proper content of TETA and OP-10. The most suitable contents of TETA and OP-10 are 15.00 and 1 wt%, respectively.

Data accessibility. The experimental data are contained in the electronic supplementary material.
Authors' contributions. Y.C. and Y.L. made substantial contributions to the conception and design of the experiment, acquisition of data, analysis and interpretation of data; Y.C. drafted the article. Y.L., C.X. and J.Z. revised it critically for important intellectual content and made final approval of the version to be published.
Competing interests. We declare we have no competing interests.
Funding. Funding in this research was offered by our own group.
Acknowledgements. We thank Dan Wang and Xiang Guo for the support offered in article drafting.

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
