## [Reviewer comments · Royal Society Open Science]

Review History

RSOS-182119.R0 (Original submission)

Review form: Reviewer 1

Is the manuscript scientifically sound in its present form?

Yes

Are the interpretations and conclusions justified by the results?

Yes

Is the language acceptable?

Yes

Is it clear how to access all supporting data?

Yes

Do you have any ethical concerns with this paper?

No

Have you any concerns about statistical analyses in this paper?

No

Recommendation?

Accept with minor revision (please list in comments)

Comments to the Author(s)

The authors present the synthesis procedure to prepare an epoxy foam using Polysilazane as a foaming agent. The manuscript is generally well written and with accuracy. However, for me this is not clear that the use of polysilazane is used for the first time as foaming agent since several publications using Polysilazane for foam synthesizes have been found. Please explain this or change to something like "To the best authors knowledge..."

Review form: Reviewer 2**Is the manuscript scientifically sound in its present form?**

No

Are the interpretations and conclusions justified by the results?

No

Is the language acceptable?

No

Is it clear how to access all supporting data?

No

Do you have any ethical concerns with this paper?

No

Have you any concerns about statistical analyses in this paper?

No

Recommendation?

Reject

Comments to the Author(s)

The article entitle "Polysilazane as a new foaming agent to prepare high strength, low density epoxy foam" was carefully reviewed.

The overall outlay and quality, novelty is not good for publication.

Hence I recommend for its rejection.

Decision letter (RSOS-182119.R0)

27-Mar-2019

Dear Dr Yongming:

Title: Polysilazane as a new foaming agent to prepare high strength, low density epoxy foam
Manuscript ID: RSOS-182119

Thank you for submitting the above manuscript to Royal Society Open Science. On behalf of the Editors and the Royal Society of Chemistry, I am pleased to inform you that your manuscript will be accepted for publication in Royal Society Open Science subject to minor revision in accordance with the referee suggestions. Please find the reviewers' comments at the end of this email.

The reviewers and handling editors have recommended publication, but also suggest some minor revisions to your manuscript. Therefore, I invite you to respond to the comments and revise your manuscript.

Because the schedule for publication is very tight, it is a condition of publication that you submit the revised version of your manuscript before 05-Apr-2019. Please note that the revision deadline will expire at 00.00am on this date. If you do not think you will be able to meet this date please let me know immediately.

Once again, thank you for submitting your manuscript to Royal Society Open Science. The chemistry content of Royal Society Open Science is published in collaboration with the Royal

Society of Chemistry. I look forward to receiving your revision. If you have any questions at all, please do not hesitate to get in touch.

Best wishes,
Dr Laura Smith
Publishing Editor, Journals

On behalf of the Subject Editor Professor Anthony Stace and the Associate Editor Professor Claire Carmalt.

RSC Associate Editor:

Comments to the Author:

Although one reviewer recommends accept with minor corrections and the other rejection I am inclined to recommend 'accept with minor correction' since the comments raised is that the novelty requires clarification. Thus, if the authors are able to describe this more clearly in the introduction this submission could be acceptable for publication. No comments were raised regarding it being unscientifically sound.

RSC Subject Editor:

Comments to the Author:

(There are no comments.)

Reviewer comments to Author:

Reviewer: 1

Comments to the Author(s)

The authors present the synthesis procedure to prepare an epoxy foam using Polysilazane as a foaming agent. The manuscript is generally well written and with accuracy. However, for me this is not clear that the use of polysilazane is used for the first time as foaming agent since several publications using Polysilazane for foam synthesizes have been found. Please explain this or change to something like "To the best authors knowledge..."

Reviewer: 2

Comments to the Author(s)

The article entitle "Polysilazane as a new foaming agent to prepare high strength, low density epoxy foam" was carefully reviewed.

The overall outlay and quality, novelty is not good for publication.

Hence I recommend for its rejection.

Author's Response to Decision Letter for (RSOS-182119.R0)

See Appendix A.

Decision letter (RSOS-182119.R1)

05-Apr-2019

Dear Dr Yongming:

Title: Polysilazane as a new foaming agent to prepare high strength, low density epoxy foam
Manuscript ID: RSOS-182119.R1

It is a pleasure to accept your manuscript in its current form for publication in Royal Society Open Science. The chemistry content of Royal Society Open Science is published in collaboration with the Royal Society of Chemistry.

On behalf of the Subject Editor Professor Anthony Stace and the Associate Editor Professor Claire Carmalt.

RSC Associate Editor
Comments to the Author:
(There are no comments.)

Reviewer(s)' Comments to Author:

Appendix A

Dear Dr Laura Smith:

Thank you for reviewing our manuscript **RSOS-182119** entitled "*Polysilazane as a new foaming agent to prepare high strength, low density epoxy foam*". I also would like to sincerely appreciate referees' effort on reviewing our manuscript and giving valuable comments.

The manuscript has been carefully checked and made minor revision according to your kind advices and referees suggestions. The respond to the comments is described in a separate sheet.

With best regards

Sincerely yours,

Yongming Luo

Reviewer #1:

The authors present the synthesis procedure to prepare an epoxy foam using Polysilazane as a foaming agent. The manuscript is generally well written and with accuracy. However, for me this is not clear that the use of polysilazane is used for the first time as foaming agent since several publications using Polysilazane for foam synthesizes have been found. Please explain this or change to something like "To the best authors knowledge..."

We made careful search in SCI and EI databases for relevant retrieval. We have only found one patent(SU578283-A), "Refractory cellular concrete mixcontg.specified alumina, phosphate binder and dimethylcyclosilazane as foaming agent to increase strength", which is related with silazane foaming agent, and no any reports about silazane as a foaming for epoxy foams was found.

Actually, there are several references about preparation of silazane foam. However, preparation of silazane foams is different from that using silazane as foaming agent. In the former, silazane was used as matrix, and formed foam in the presence of other foaming agent. In our work, epoxy resin was matrix, which formed epoxy foam in the presence of silazane foaming agent. There is no report about using silazane as foaming agent to prepare epoxy foam before our work. So we think our manuscript "Polysilazane as a new foaming agent to prepare high strength, low density epoxy foam" has novelty.